# Impacts of Side Effects to BNT162b2 and the First Dose of ChAdOx1 Anti-SARS-CoV-2 Vaccination on Work Productivity, the Need for Medical Attention, and Vaccine Acceptance: A Multicenter Survey on Healthcare Workers in Referral Teaching Hospitals in the Republic of Korea

**DOI:** 10.3390/vaccines9060648

**Published:** 2021-06-14

**Authors:** Tark Kim, Se Yoon Park, Shinae Yu, Jung Wan Park, Eunjung Lee, Min Hyok Jeon, Tae Hyong Kim, Eun Ju Choo

**Affiliations:** 1Division of Infectious Diseases, Department of Internal Medicine, Soonchunhyang University Bucheon Hospital, Soonchunhyang University College of Medicine, Bucheon 14584, Korea; ktocc@schmc.ac.kr; 2Division of Infectious Diseases, Department of Internal Medicine, Soonchunhyang University Seoul Hospital, Soonchunhyang University College of Medicine, Seoul 04401, Korea; sypark@schmc.ac.kr (S.Y.P.); shegets@schmc.ac.kr (E.L.); geuncom@schmc.ac.kr (T.H.K.); 3Division of Infectious Diseases, Department of Internal Medicine, Soonchunhyang University Cheonan Hospital, Soonchunhyang University College of Medicine, Cheonan 31151, Korea; zolzoly@schmc.ac.kr (S.Y.); splendidmagic@schmc.ac.kr (J.W.P.); yacsog@schmc.ac.kr (M.H.J.)

**Keywords:** COVID-19, SARS-CoV-2, vaccination, side effect

## Abstract

To establish a successful anti-SARS-CoV-2 vaccination strategy, it is necessary to take possible tradeoffs into account. We conducted a survey on vaccinated healthcare workers (HCWs) inthree referral teaching hospitals in the Republic of Korea. We investigated the frequency of vaccination side effects (SEs), the impact on their work productivity, the need for medical attention, and vaccine acceptance. Three groups of HCWs were surveyed: 1406 who had received the first dose of BNT162b2 (BNT162b2#1), 1168 who had received the second dose of BNT162b2 (BNT162b2#2), and 1679 who had received the first dose of ChAdOx1 (ChAdOx1#1). More SEs and impact on work productivity were reported in ChAdOx1#1 than in the other two groups. However, among individuals aged ≥40 years, no significant difference of absence from work was found between ChAdOx1#1 and BN162b2#2 (4.4%, 31/699 vs. 3.0%, 12/405; *p* = 0.26), and none were hospitalized. Older HCWs in ChAdOx1#1 showed intention to receive the second dose of the vaccine. Although the incidence of SEs and their impacts were greater after the first dose of ChAdOx1 than BNT162b2 in young people, significant impact of SEs seemed to be rare in individuals aged ≥40 years, regardless of the vaccine they received.

## 1. Introduction

Severe acute respiratory syndrome coronavirus 2 (SARS-COV-2) was first detectedin Wuhan, China in December 2019 [1]. It has resulted in 141,057,106 infections and 3,015,043 deaths worldwide as of 18 April 2021 [2]. The efficacy of the current therapeuctic regimens for SARS-CoV-2 infection is still limited. Nonpharamaceutical intervention such as social distancing to contain the epidemic is accompanied by socio-economic collateral damage. Vaccination against SARS-CoV-2 will be the the best way to control it.

With huge investments and dedicated effort by scientists, vaccines against SARS-CoV-2 have been commercialized less than a year after the declaration of the SARS-CoV-2 pandemic [3]. Vaccines that are currently in use have shown their efficacy and safety in phase III trials [4,5]. Realworld data from Scotland and Israel, as countries having the highest vaccination rates, have also proven the effectiveness of mRNA vaccines and a virus-vector vaccine [6,7].

Nevertheless, people have concerns about the safety of the vaccines based on new platforms. Especially, ChAdOx1 adenovirus-vector vaccine developed by Astrazeneca-Oxford University is at the center of controversy due to vaccine-induced prothrombotic immune thrombocytopenia, which is a very rare side effect reported in five out of 130,000 vaccinated individuals [8]. Although the European Medicines Agency reported that the benefit of the vaccine overwhelmingly exceeds the risk [9], this concern may reduce vaccine acceptance [10]. In addition, systemic side effects such as fever and myalgia can decrease work productivity and increase the need for medical attention. Therefore, it is important to understand the frequency of side effects and their effects on daily life in order to establish a successful vaccination strategy.

In the Republic of Korea, healthcare workers (HCWs) in referral hospitals were classified as the first the vaccination priority group. They received either the Pfizer-BioNTech BNT162b2 mRNA vaccine or the ChAdOx1 vaccine. In the present study, we investigated the frequency of anti-SARS-CoV-2 vaccine side effects, their impact on work productivity, the subsequent need for medical attention, and vaccine acceptance among the vaccinated HCWs in three referral teaching hospitals.

## 2. Subjects and Methods

### 2.1. Study Design &Population

This study included HCWs in three referral teaching hospitals with 725 beds, 879 beds, and 899 beds, located in Seoul, Bucheon, and Cheonan, in the Republic of Korea. HCWs were classified as candidates for receiving anti-SARS-CoV-2 vaccination in the first quarter of year 2021 based on the vaccine priority strategy in the Repblic of Korea. ChAdOx1 vaccine was allocated for them, while BNT162b2 vaccine was allocated for HCWs at dedicated hospitals for COVID-19. The number of candidates for BNT162b2 vaccine was as many as ten times the number of dedicated beds for severe COVID-19 in other referral teaching hospitals. 

### 2.2. Survey

The questionnaires were in form of Google Forms (Google, San Francisco, CA, USA). Links to questionnaires were sent to the vaccinated HCWs by SNS, e-mail, and text messages. Vaccination and survey timelines are illustrated in Figure 1. Questionnaires for the second dose of anti-SARS-CoV-2 vaccine were only sent to those who had received the second dose of BNT162b2 (Figure 1), since the second dose of ChAdOx1 was not yet due (after 12 weeks from the first dose). The survey for the second dose vaccination was conducted independently, regardless the first dose survey response, because personal identifiable information was not included in the survey. The maximum duration from the vaccination to survey date for the first dose of BNT162b2 was 41 days, and the minimum duration for the second dose of BNT162b2 was 5 days (Figure 1). The survey included an explanation of the research purpose, and vaccinated HCWs responded on consent. The questionnaire included demographics (age group, gender), the name of the hospital, the type of job, type of vaccination, previous history of confirmed SARS-CoV-2 infection, pre- and post-vaccination medications, need for medical attention, and impact on work productivity (vacation or holidays, impaired work performance, and absence from work). Those who answered that they visited an out-patient department, emergency department, or hospitalized due to any side effects were classified as having “needed medical intention”. Those who responded that they went to work but could not work normally due to any side effects after vaccination were classified as “work performance impaired”. To simplify the questionnaire as much as possible, the time of onset, duration, and severity for each side effect were not included. To evaluate vaccine acceptance in the questionnaire, the HCW were asked if they would receive a second dose of the vaccination after the first dose.

### 2.3. Statistics

All statistical analyses were performed using the SPSS Statistics version 26.0 (SPSS, Chicago, IL, USA) and the MedCalc version 19.3 (MedCalc Software Ltd., Ostend, Belgium). Categorical variables were compared using the Chi-squared test or the Fisher’s exact test. Trends of a significant increase or decrease in the frequency of side effects across age groups were estimated by linear-by-linear association. Binary logistic regression was used to identify risk factors associated with impaired work performance or absence from work resulting from anti-SARS-CoV-2 vaccination side effects. Among variables of gender, age group, previous confirmed SARS-CoV-2 infection, history of allergy or anaphylaxis, and prevaccination medication, those with statistical significance at the 5% level in the univariate analysis were chosen for the multivariate analysis. There were few people who had impaired work performance in the group that received the first dose of BNT162b2 (the BNT162b2 #1 group) for multivariate analysis, and only responses of the group that received the second dose of BNT162b2 (the BNT162b2 #2 group) and the first dose of ChAdOx1 (the ChAdOx1 #1 group) were included for the evaluation of the risk factors. Responses of having a vacation or holiday after vaccination were excluded from the analysis. All tests were two-tailed and differences were considered significant if *p* < 0.05.

## 3. Results

The number of vaccine candidates, vaccination rates, and the survey response rates are illustrated in Figure 1. Among respondents who received the second dose of BNT162b2, 83.1% (971/1168) participated in the survey after receiving the first dose of the vaccine. Demographics of the respondents according to the vaccine type and dose are described in Table 1. Females accounted for 75.6% (3216). The most common age group was 20–29 years of age (34.0%), followed by 30–39 (27.8%). Among the respondents, 157 (3.7%) were more than 60 years old. The most common type of job was for nurses (44.9%), followed by doctors (14.1%), medical technicians (20.5%), and administrative staff (9.3%). Sixty (1.4%) respondents had had previous SARS-CoV-2 infecton. A previous history of allergy to drugs, food, or vaccine was reported by 8.7% (368) of respondents, and a history of anaphylaxis was reported in 1.0% (41) of the respondents.

Differences in the vaccine side effects frequencies and their impacts on need for medical attention and work productivity after anti-SARS-CoV-2 vaccination are shown in Table 1 and Figure 2. Many respondents complained of local adverse reactions, regardless of the vaccine type. General myalgia was the most common systemic side effect, followed by febrile sensations, chills, fatigue, headache, arthralgia, dizziness, and nausea. These side effects occurred within 12hrs after vaccination in 25.4% of the HCWs in the ChAdOx1 #1 group, while they occurred within 12hrs in 70.5% of HCWs in the BNT162b2 #1 group (*p* < 0.01) and 56.7% of the HCWs in the BNT162b2 #2 group (*p* < 0.01). Although most of these side effects disappeared in less than 48 hrs, 29.7% of the HCWs in the ChAdOx1 #1 group complained side effects for more than 48hrs. which was longer than in the BNT162b2 #1 group (14.6%; *p* < 0.01) and the BNT162b2 #2 (22.9%; *p* < 0.01). Except for local tenderness, dizziness, pruritus, and rash, the number of those who experienced side effects in the BNT162b2 #2 group was greater than that for those in the BNT162b2 #1 group. Except for local tenderness, local edema, vomit, dyspnea, and rash, the incidence rates in the ChAdOx1 #1 group were higher than those in the BNT162b2 #2 group. Likewise, HCWs in the ChAdOx1 #1 group the number of the HCWs who took medicine after vaccination (81.2%) was the highest, followed by those in the BNT162b2 #2 group (74.1%) and the BNT162b2 #1group (32.9%). More HCWs in the ChAdOx1 #1 group had impaired work performance or were abscent from work due to vaccine side effects compared to those in the BNT162b2 #2 group (27.0% in the ChADOx1 #1group vs. 21.6% in the BNT162b2 #2 group; *p* < 0.01). One hundred and three (8.5%) HCWs in the ChADOx1 #1 group and 38 (3.3%) HCWs in the BNT162b2 #2 group visited the out-patient department or the emergency department (*p* < 0.01). At the time of the survey after the first dose, only 78.8% of the respondents in the ChAdOx1 #1 group and 98.9% of respondents in the BNT162b2 #1 group answered that they would receive the second dose. Reasons for declining the second dose in both the BNT162b2 #1 group and the ChAdOx1 #1 group were side effects to the first dose (30.7% (112)) and concern about side effects (29.0% (108)). In the ChAdOx1 #1 group aged <50 year-old, respondents who declined the second dose vaccination more commonly experienced a local erythema/heating sensation, local swelling, chills, fatigure, and headache than those who wanted to receive the second dose (Appendix A).

Differences in side effects to vaccines and their impacts on the need for medical attention, and work productivity between the ChAdOx1 #1 group and the BNT162b2 #2 group according to age groups are shown in Table 2 and Figure 3. Older groups experienced fewer side effects, regardless of the vaccine type (Appendix A). Compared to the BNT162b2 #2 group, the frequency of side effects and their impact on work productivity and the need for medical attention was significantly high in the ChAdOx1 #1 20–29 year-old group and 30–39 year-old group, while there were no statistically significant differences the 60≥ year-old group. Although in the 40–49 year-old group, more HCWs in the ChAdOx1 #1 group complained of febrile sensations, chills, headache, arthralgia, dizziness, and pruritus, impaired work performance or absence from work and the need for medical attention, this was not statistically different compared to those in the BNT162b2 #2 group. In the 50–59 year-old group, there were more complaints on febrile sensationsand pruritus among the HCWs in the ChAdOx1 #1 group compared to those in the BNT162b2 #2 group. Visits to an out-patient or emergency department were only necessary for 2.0% (8/405) of the HCWs in the BNT162b2 #2 group and 5.1% (34/699) of the HCWs in the ChAdOx1 #1 group for those aged over 40 years. In the ChAdOx1 #1 group, 95.9% of the HCWs aged ≥60 years and 76.1% of the HCWs aged <40 years showed an intention to receive the second dose.

Risk factors for impaired work performance or absence from work in both the ChAdOx1 #1 group and the BNT162b2 #2 group are shown in Table 3. Female gender (adjusted odd ratio [aOR]:1.38, 95% confidence interval [95% CI]:1.11–1.72; *p* < 0.01), the first dose of ChAdOx1 (aOR:2.05, 95% CI:1.69–2.49; *p* < 0.01), and age less than 40 years (aOR: 2.54, 95% CI: 2.06–3.12; *p* < 0.01) were siginificantly associated with impaired work performance or absence from work after vaccination. Prevaccination medication did not decrease the chances of impaired work performance or absence from work after vaccination.

## 4. Discussion

Based on the present study, the frequency of side effects, their impact on productivity and need for medical attention were highest after the first dose of ChAdOx1 vaccination, followed by the second dose of BNT162b2 and the first dose of BNT162b2. However, the side effects after the second dose of ChAdOx1 were not surveyed. Older HCWs showed fewer side effects, impaired work performance or absence from work, or visits to the out-patient department or the emergency department after the vaccination. Among the HCWs aged over 40 years, less than 5% needed medical attention after either ChAdOx1 or BNT162b2 vaccination. These results are expected to provide important data for establishing vaccine strategies and responding to side effects after anti-SARS-CoV-2 vaccination.

Previous reports have shown similar findings. A COVID-19 vaccine safety update published by the Advisory Committee on Immunization Practices [11] reported that local tenderness (74.8%) was the most common side effect, followed by fatigue (50.0%), headache (41.9%), myalgia (41.6%), chills (26.7%), fever (25.2%), injection site swelling (26.7%), arthralgia (21.2%), and nausea (13.9%) after the second dose of BNT162b2 vaccine. On the other hand, local tenderness (67.7%), fatigue (28.6%), headache (25.6%), myalgia (17.2%), chills (7.0%), fever (7.5%), injection site swelling (6.8%), arthralgia (7.1%) and nausea (7.0%) (11) were less frequently reported in the group receiving the first dose of BNT162b2 [11]. In a phase 2/3 trial of the ChAdOx1vaccine, local tenderness (76%), fatigue (76%), headache (65%), general myalgia (53%), chills (35%), arthralgia (33%), nausea (26%), and febrile sensations (24%) were common side effects in participants aged 18–55 year-old, while local tenderness (49%), fatigue (41%), headache (41%), general myalgia (18%), chills (4%), arthralgia (14%), nausea (8%), and febrile sensations(0%) were less frequently reported in participants aged >70 years [12]. However, since these data were obtained under different conditions, it is not reasonable to directly compare these differences between BNT162b2 and ChAdOx1. In addition, the age group of participants under 55 year-old was not subdivided [12]. There have been a few real-world survey studies on the side effects ofSARS-CoV-2 vaccination [13,14,15,16,17,18]. However, the previous studies cited here included only those receiving the first dose of BNT162b2 [13,14], a small number of BNT162b2 [15,16,17], and the mRNA 1273 vaccine produced by Moderna [18]. Comendably, our study included a large number of respondents who received the second dose of BNT162b2 and the first dose of ChAdOx1 under the same conditions.

Our findings on the side effects and their impacts on work productivity, need to seek medical attention, and vaccine acceptance according to age group and vaccine type provide important suggestions for anti-SARS-CoV-2 vaccine strategies. Vaccination for those aged ≥60 years is important to decrease the mortality due to COVID-19, because old age is one of the most important mortality risk factors [19]. Our study indicated that both ChAdOx1 and BNT162b2 are well tolerated by individualsaged ≥40 years. Vaccination for younger persons is important to prevent the transmission of SARS-CoV-2 infection and protect essential facilities, although the benefit on mortality and morbidity might be less for younger persons. However, the high frequency of side effects and their impacts on work productivity, need for medical attention and vaccine accceptance among the young individuals in our study raises important concerns. Indeed, based on our results, there were significantly high numbers of side effects complaints among respondents who declined the second dose of ChAdOx1 vaccine (Appendix A). Similarly, Rief W also identified the side effects that affected vaccine acceptance [10]. Based on our findings, if possible, it would be better to give BNT162b2 to those aged <40 years and not ChAdOx1, although a previous survey study showed that 12.3% of HCWs took time off from work after BNT162b2 vaccination [13]. In addition, two or three days of vacation after vaccination should be considered for young people, and should also be measured to caution against staff shortages in case many staff members require sick leave after vaccination. We believe that preparedness and appropriate responses towards anti-SARS-CoV-2 vaccine side effects will help increase vaccine acceptance.

This survey study has several limitations. First, it was a survey study, and therefore, might have been affected by memory biases. To reduce recall bias, the survey was requested shortly after vaccination (Figure 1). Those who experienced side effects might have responded more actively than those who did not. We hope that more accurate results will be obtained with an established active pharmacovigilance system through which all vaccinated people should respond. A passive pharmacovigilance system using IT technologies would also be a good means of monitoring side effects [20]. Second, the onset time and duration of different side effects were not investigated. However, this study did not aim to investigate the severity and duration of different side effects, and it is difficult to determine specific side effects that affected the work productivity, need for medical attention, and vaccine acceptance. There was also a concern on lowering the response rate in case of a too detailed questionnaire. Third, the variable for impaired work performance was not specifically quantified, and specific reasons for medical attention were not investigated, even though these impacts were due to side effects. The high rate of subsequent need for medical attention can be attributed to the Korean health system, which has easy access to medical care and low medical costs. Furthermore, there was no specific diagnosis of side effects such as thrombosis. Fourth, age and job distributions were not the same between BNT162b2 and ChAdOx1 groups. Different age distribution did not affect the conclusion, because older HCWs in ChAdOx1 group had fewer SEs. However, there is a possibility of bias inthe results on work productivity due to different job distribution. Finally, the data on the side effects after the second dose of ChAdOx1 were not included in this study. In phase 2/3 of clinical trials, a lower frequency of side effects was reported after the ChAdOx1 boostershot compared to the primer [12]. For this reason, we think that it is reasonable to compare the side effects of the second dose of BNT162b2 to the first dose of ChADOx1. Similar to our study, this was successfully done in a recent study, where the side effects between the second dose of BNT162b2 and the first dose of ChADOx1 were compared [20]. Furthermore, since a delayed ChAdOx1 boosting shot is recommended, the delay makes it difficult to conduct timely studies. Nevertheless, the lack of analysis on the side effects after the second dose of ChAdOx1 should be taken into account in interpreting our results. To clearly elucidate the impact of anti-SARS-CoV-2 vaccine on work productivity, the need for medical attention, and vaccine acceptance, further studies, including ananalysis on the side effects after the second dose of ChAdOx1, are required.

In conclusion, the incidences of side effects were higher among younger people than in older individuals. Consequently, their work productivity was more affected by BNT162b2 and ChOxAd1 side effects. However, the impacts of side effects after vaccination with BNT162b2 and the first dose of ChOxAd1 were similar and tolerable in people aged ≥40 years. These findings suggest that to increase vaccine acceptance, it may be advisable to vaccinate people below 40 years with BNT162b2 and not ChOxAd1. These results are expected to provide important data for establishing vaccine strategies and responding to side effects after anti-SARS-CoV-2 vaccination

## Figures and Tables

**Figure 1 vaccines-09-00648-f001:**
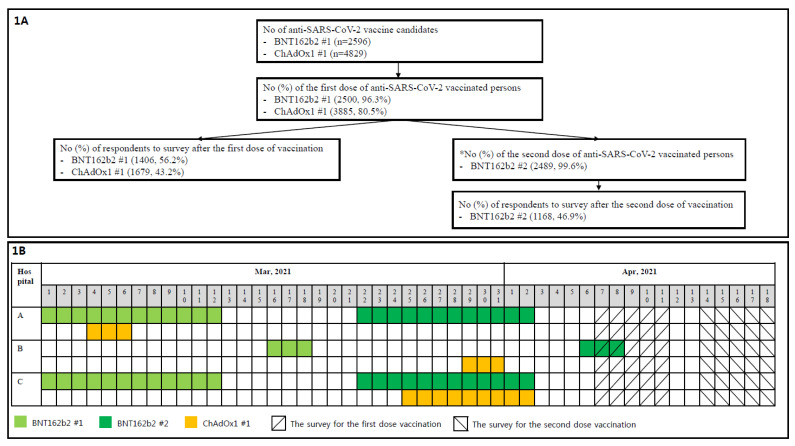
Study design. (**A**) is the study flowchart. (**B**) shows the vaccination and survey timelines. ChAdOx1 #1, the first dose of ChAdOx1; BNT162b2 #1, the first dose of BNT162b2; BNT162b2 #2, the second dose of BNT162b2. * The schedule of the second dose of ChAdOx1 was not reached during the study period.

**Figure 2 vaccines-09-00648-f002:**
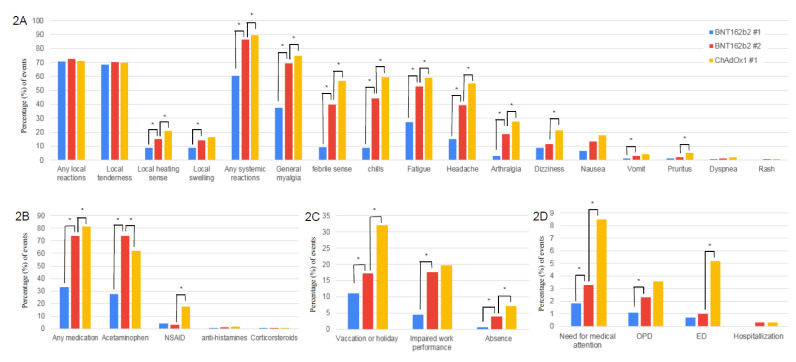
Side effectsand their impacts on post-vaccination medication, work, and medical use after anti-SARS-CoV-2 vaccination according to vaccine dose and type. (**A**) shows side effects. (**B**) shows post-vaccination medication. (**C**) shows impact on work. (**D**) shows medical use after vaccination. ChAdOx1 #1, the first dose of ChAdOx1; BNT162b2 #1, the first dose of BNT162b2; BNT162b2 #2, the second dose of BNT162b2; NSAID, nonsteroid anti-inflammtory drug; ED, Emergency department; OPD, out-patient department. * *p*-value < 0.05 when comparing frequencies of events according to vaccine type and dose.

**Figure 3 vaccines-09-00648-f003:**
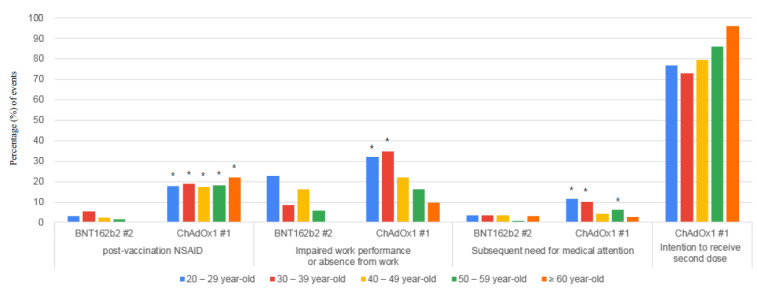
Side effects and their impactson post-vaccination medication, work, and medical use after the first dose of ChAdOx1and the second dose of BNT162b2 vaccination according toage groups. ChAdOx1 #1, the first dose of ChAdOx1; BNT162b2 #2, the second dose of BNT162b2; NSAID, nonsteroid anti-inflammtory drug. * *p*-value < 0.05 when comparing frequencies of events between BNT162b2 #2 and ChAdOx1 #1 groups for those belonging to the same age group.

**Table 1 vaccines-09-00648-t001:** Side effects and their impact on work productivity and need for medical attention after BNT162b2 and ChAdOx1 anti-SARS-CoV-2 vaccination.

	BNT162b2 #1 (*n* = 1406)	BNT162b2 #2 (*n* = 1168)	ChAdOx1 #1 (*n* = 1679)	P1	P2	P3
Gender, male	346 (24.6)	292 (25.0)	399 (23.8)	0.82	0.61	0.45
Age						
20–29 year-old	445 (31.7)	390 (33.4)	613 (36.5)	0.35	<0.01	0.09
30–39 year-old	444 (31.6)	373 (31.9)	367 (21.9)	0.87	<0.01	<0.01
40–49 year-old	307 (21.9)	236 (20.2)	360 (21.4)	0.33	0.79	0.45
50–59 year-old	160 (11.4)	136 (11.6)	265 (15.8)	0.85	<0.01	<0.01
≥60 year-old	50 (3.6)	33 (2.8)	74 (4.4)	0.31	0.03	0.27
Job						
Doctor	216 (15.4)	200 (17.1)	182 (10.8)	0.24	<0.01	<0.01
Nurse	684 (48.6)	571 (48.9)	654 (39.0)	0.91	<0.01	<0.01
Medical techinican	258 (16.3)	204 (17.5)	409 (24.4)	0.57	<0.01	<0.01
Administrative	110 (7.8)	97 (8.3)	187 (11.1)	0.66	<0.01	<0.01
Others	138 (9.8)	96 (8.2)	247 (14.7)	0.17	<0.01	<0.01
Previous confirmed SARS-CoV-2 infection	22 (1.6)	27 (2.3)	11 (0.7)	0.19	0.02	<0.01
History of allergy to						
Any drug or food	128 (9.1)	94 (8.0)	97 (5.8)	0.36	<0.01	0.02
Any vaccine	16 (1.1)	13 (1.1)	20 (1.2)	1.00	1.00	1.00
History of anaphylaxis to						
Any drug or food	9 (0.6)	9 (0.8)	15 (0.9)	0.81	0.54	0.84
Any vaccine	1 (0.1)	0	8 (0.5)	1.00	0.04	0.02
Local reactions	996 (70.8)	849 (72.7)	1195 (71.2)	0.31	0.84	0.4
Local tenderness	861 (68.3)	822 (70.4)	1168 (69.6)	0.28	0.48	0.65
Local erythema/heating sensation	123 (8.7)	176 (15.1)	349 (20.8)	<0.01	<0.01	<0.01
Local edema	125 (8.9)	166 (14.2)	273 (16.3)	<0.01	<0.01	0.14
Systemic reactions	850 (60.5)	1010 (86.5)	1501 (89.4)	<0.01	<0.01	0.02
General myalgia	525 (37.3)	812 (69.5)	1252 (74.6)	<0.01	<0.01	<0.01
Febrile sensations	131 (9.3)	465 (39.8)	954 (56.8)	<0.01	<0.01	<0.01
Chills	125 (8.9)	515 (44.1)	997 (59.4)	<0.01	<0.01	<0.01
Fatigue	379 (27.0)	618 (52.9)	990 (59.0)	<0.01	<0.01	<0.01
Headache	213 (15.1)	457 (39.1)	924 (55.0)	<0.01	<0.01	<0.01
Arthralgia	41 (2.9)	219 (18.8)	464 (27.6)	<0.01	<0.01	<0.01
Dizziness	122 (8.7)	136 (11.6)	361 (21.5)	0.12	<0.01	<0.01
Nausea	80 (6.4)	155 (13.3)	298 (17.7)	<0.01	<0.01	<0.01
Vomit	20 (1.4)	35 (3.0)	70 (4.2)	<0.01	<0.01	0.11
Pruritus	18 (1.4)	26 (2.2)	85 (5.1)	0.10	<0.01	<0.01
Dyspnea	8 (0.6)	12 (1.0)	32 (1.9)	0.26	<0.01	0.07
Rash	3 (0.2)	8 (0.7)	15 (0.9)	0.08	0.02	0.67
Onset time of side effects after vaccination						
<3 h	286 (20.3)	126 (10.8)	111 (6.6)	<0.01	<0.01	<0.01
3–6 h	273 (19.4)	181 (15.5)	125 (7.4)	<0.01	<0.01	<0.01
6–12 h	434 (30.9)	355 (30.4)	190 (11.3)	0.80	<0.01	<0.01
12–24 h	280 (19.9)	349 (29.9)	792 (47.2)	<0.01	<0.01	<0.01
24–48 h	78 (5.5)	94 (8.0)	374 (22.3)	0.01	<0.01	<0.01
≥48 h	33 (2.3)	7 (0.6)	77 (4.6)	<0.01	<0.01	<0.01
Duration of side effects						
<24 h	544 (38.7)	306 (26.2)	359 (21.4)	<0.01	<0.01	<0.01
24–48 h	635 (45.2)	540 (46.2)	710 (42.4)	0.61	0.11	0.04
48–72 h	155 (11.0)	196 (16.2)	335 (20.0)	<0.01	<0.01	0.04
72 h–5 days	26 (1.9)	51 (4.4)	98 (5.5)	<0.01	<0.01	0.09
≥5 days	24 (1.7)	20 (1.7)	65 (3.5)	0.88	<0.01	<0.01
Prevaccination medication	110 (7.8)	209 (17.9)	441 (26.3)	<0.01	<0.01	<0.01
Acetaminophen	87 (6.2)	187 (16.0)	379 (22.6)	<0.01	<0.01	<0.01
NSAID	15 (1.1)	22 (1.9)	55 (3.2)	0.10	<0.01	0.03
Anti-histamine	8 (0.6)	9 (0.5)	6 (0.4)	0.63	0.43	0.19
Corticosteroids	0	0	1 (0.1)	1.00	1.00	1.00
Post-vaccination medication	462 (32.9)	866 (74.1)	1364 (81.2)	<0.01	<0.01	<0.01
Acetaminophen	391 (27.8)	861 (73.7)	1040 (61.9)	<0.01	<0.01	<0.01
NSAID	60 (4.3)	40 (3.4)	298 (17.7)	0.31	<0.01	<0.01
Anti-histamine	12 (0.9)	12 (1.0)	30 (1.8)	0.68	0.03	0.11
Corticosteroids	1 (0.1)	3 (0.3)	4 (0.2)	0.34	0.38	1.00
Impact on work productivity						
Vacation or holiday	154 (11.0)	201 (17.2)	539 (32.1)	<0.01	<0.01	<0.01
Impaired work performance	63 (4.5)	207(17.7)	331 (19.7)	<0.01	<0.01	0.19
Absence from work	8 (0.6)	45 (3.9)	123 (7.3)	<0.01	<0.01	<0.01
Subsequent need for medical attention	26 (1.8)	38 (3.3)	143 (8.5)	0.03	<0.01	<0.01
OPD visit	16 (1.1)	27 (2.3)	60 (3.6)	0.03	<0.01	0.06
ED visit	10 (0.7)	12 (1.0)	88 (5.2)	0.40	<0.01	<0.01
Hospitalization	0	3 (0.3)	5 (0.3)	0.09	0.07	1.00
Intend to second dose	1390 (98.9)	NA	1323 (78.8)	NA	<0.01	NA

Data are presented as numbers (%) of respondants. P1, *p* value between BNT162b2 #1 and BNT162b2 #2; P2, *p* value between BNT162b2 #1 and ChAdOx1 #1; P3, *p* value between BNT162b2 #2 and ChAdOx1 #1. BNT162b2 #1, the first dose of BNT162b2; BNT162b2 #2, the second dose of BNT162b2; ChAdOx1 #1, the first dose of ChAdOx1; NSAID, nonsteroid anti-inflammatory drug; OPD, out-patient department; ED, emergency department; NA, not available.

**Table 2 vaccines-09-00648-t002:** Side effects and their impact on work productivity and need for medical attention after the second dose of BNT162b2 and the first dose of ChAdOx1 vaccination according to age groups.

	20–29 Year-Old			30–39 Year-Old			40–49 Year-Old			50–59 Year-Old			60≥ Year-Old		
	BNT162b2 #2 (*n* = 390)	ChAdOx1 #1 (*n* = 613)	*p*-Value	BNT162b2 #2 (*n* = 373)	ChAdOx1 #1 (*n* = 367)	*p*-Value	BNT162b2 #2 (*n* = 236)	ChAdOx1 #1 (*n* = 360)	*p*-Value	BNT162b2 #2 (*n* = 136)	ChAdOx1 #1 (*n* = 265)	*p*-Value	BNT162b2 #2 (*n* = 33)	ChAdOx1 #1 (*n* = 74)	*p*-Value
Local reactions	290 (74.4)	496 (79.1)	0.09	289 (77.5)	279 (76.0)	0.66	158 (66.9)	247 (68.6)	0.72	92 (67.6)	153 (57.7)	0.07	20 (60.6)	31 (41.9)	0.09
Local tenderness	277 (71.0)	476 (77.7)	0.02	284 (76.1)	278 (25.7)	0.93	151 (64.0)	239 (68.4)	0.60	91 (66.9)	147 (55.5)	0.03	19 (57.6)	28 (37.8)	0.06
Local erythema/heating sensation	77 (19.7)	157 (25.6)	0.03	59 (15.8)	77 (21.0)	0.07	21 (8.9)	70 (19.4)	<0.01	15 (11.0)	39 (14.7)	0.36	4 (12.1)	6 (8.1)	0.49
Local edema	71 (18.2)	120 (19.6)	0.62	46 (12.3)	60 (16.3)	0.14	30 (12.7)	51 (14.2)	0.63	15 (11.0)	35 (13.2)	0.63	4 (12.1)	7 (9.5)	0.74
Systemic reactions	340 (87.2)	582 (94.9)	<0.01	338 (90.6)	346 (94.3)	0.07	203 (80.6)	319 (88.6)	0.38	110 (80.9)	211 (79.6)	0.78	19 (57.6)	41 (55.4)	1.00
General myalgia	261 (66.9)	512 (83.5)	<0.01	281 (75.3)	296 (80.7)	0.09	165 (69.9)	262 (72.8)	0.46	90 (66.2)	160 (60.4)	0.28	15 (45.5)	22 (29.7)	0.13
Febrile sensation	181 (46.4)	465 (75.9)	<0.01	184 (49.3)	236 (64.3)	<0.01	67 (28.4)	164 (45.6)	<0.01	29 (21.3)	82 (30.9)	0.04	4 (12.1)	7 (9.5)	0.74
Chills	175 (44.9)	449 (73.2)	<0.01	200 (53.6)	251 (68.4)	<0.01	92 (39.0)	182 (50.6)	<0.01	43 (31.8)	100 (37.7)	0.27	5 (15.2)	15 (20.2)	0.60
Fatigue	221 (56.7)	393 (64.1)	0.02	203 (54.4)	254 (69.2)	<0.01	120 (50.8)	189 (52.5)	0.74	62 (45.6)	133 (50.2)	0.40	12 (36.4)	21 (28.4)	0.50
Headache	184 (47.2)	414 (67.5)	<0.01	152 (40.8)	238 (64.9)	<0.01	83 (35.2)	166 (45.1)	0.01	35 (25.7)	92 (34.7)	0.07	3 (9.1)	14 (19.9)	0.26
Arthralgia	58 (14.9)	172 (28.1)	<0.01	83 (22.3)	102 (27.8)	0.09	45 (19.1)	115 (31.9)	<0.01	29 (21.3)	66 (24.9)	0.46	4 (12.1)	9 (12.2)	1.00
Dizziness	57 (14.6)	168 (27.4)	<0.01	43 (11.5)	78 (21.3)	<0.01	21 (8.9)	73 (20.3)	<0.01	15 (11.0)	36 (13.8)	0.53	0	6 (8.1)	0.17
Nausea	61 (15.6)	141 (23.0)	<0.01	60 (16.1)	83 (22.6)	0.03	25 (10.6)	55 (15.3)	0.11	8 (5.9)	17 (6.4)	1.00	1 (3.0)	2 (2.7)	1.00
Vomit	11 (2.8)	32 (5.2)	0.08	16 (4.3)	21 (5.7)	0.40	5 (2.1)	10 (2.8)	0.79	3 (2.2)	6 (2.3)	1.00	0	1 (1.4)	1.00
Pruritus	8 (2.1)	27 (4.4)	0.05	12 (3.2)	23 (6.3)	0.06	5 (2.1)	25 (6.9)	0.01	0	9 (3.4)	0.03	1 (3.0)	1 (1.4)	0.52
Dyspnea	4 (1.0)	12 (2.0)	0.31	3 (0.8)	12 (3.3)	0.02	1 (0.4)	5 (1.4)	0.41	2 (1.5)	3 (1.1)	1.00	2 (6.1)	0	0.09
Rash	4 (1.0)	6 (1.0)	1.00	4 (1.1)	4 (1.1)	1.00	0	1 (0.3)	1.00	0	3 (1.1)	0.55	0	1 (1.4)	1.00
Impact on work productivity															
Vaccation or holiday	63 (16.2)	207 (33.8)	<0.01	93 (24.9)	121 (33.0)	<0.01	51 (21.6)	137 (38.1)	<0.01	20 (14.7)	67 (25.3)	0.02	0	7 (9.5)	0.10
Impaired work performance	73 (18.7)	138 (22.5)	0.15	17 (4.6)	95 (25.9)	<0.01	30 (12.7)	63 (17.5)	0.13	11 (8.1)	30 (11.3)	0.39	0	5 (6.8)	0.32
Absence from work	16 (4.1)	59 (9.6)	<0.01	15 (4.0)	33 (9.0)	<0.01	8 (3.4)	16 (4.4)	0.67	4 (2.9)	13 (4.9)	0.44	0	2 (2.7)	1.00
Need for medical attention	13 (3.3)	72 (11.7)	<0.01	13 (3.5)	37 (10.1)	<0.01	8 (3.4)	16 (4.4)	0.67	1 (0.7)	16 (6.0)	0.02	1 (3.0)	2 (2.7)	1.00
OPD visit	8 (2.1)	29 (4.7)	0.04	2 (0.5)	17 (4.6)	0.46	4 (1.7)	4 (1.1)	0.72	1 (0.7)	8 (3.0)	0.28	1 (3.0)	2 (2.7)	1.00
ED visit	5 (1.2)	48 (7.8)	<0.01	3 (0.8)	19 (5.2)	<0.01	5 (2.1)	12 (3.3)	0.46	0	9 (3.4)	0.03	0	0	NA
Hospitalization	0	4 (0.7)	0.16	3 (0.8)	1 (0.3)	0.82	0	0	NA	0	0	NA	0	0	NA

Data are presented as numbers (%) of respondents.BNT162b2 #2, the second dose of BNT162b2; ChAdOx1 #1, the first dose of ChAdOx1; OPD, out-patient department; ED, emergency department; NA, not available.

**Table 3 vaccines-09-00648-t003:** Risk factors for impaired work performance or absence from work due to side effectsafter the second dose of BNT162b2 and the first dose of ChAdOx1 vaccination.

	Impaired Work Performance or Absence on Work (*n* = 706)	No Impact (*n* = 1401)	Odds Ratio (95% CI)	Adjusted Odds Ratio (95% CI)
Gender, female vs. male	549 (77.8)	969 (69.2)	1.56 (1.26–1.92)	1.38 (1.11–1.72)
Vaccine				
ChAdOx1 #1 vs. BNT162b2 #2	454 (64.3)	686 (49.0)	1.88 (1.56–2.26)	2.05 (1.69–2.49)
Age				
20–29 year-old vs. ≥30 year-old	286 (40.5)	447 (31.9)	1.45 (1.21–1.75)	
20–39 year-old vs. ≥40 year-old	524 (74.2)	761 (54.3)	2.42 (1.99–2.95)	2.54 (2.06–3.12)
20–49 year-old vs. ≥50 year-old	641 (90.8)	1052 (75.1)	3.72 (2.47–4.34)	
20–59 year-old vs.≥60 year-old	699 (99.0)	1308 (93.4)	7.10 (3.28–15.39)	
Previous confirmed SARS-CoV-2 infection vs. No	5 (0.7)	27 (1.9)	0.36 (0.14–0.95)	0.37 (0.14–1.01)
History of allergy				
To any drug or food vs. No	52 (7.4)	78 (5.6)	1.35 (0.94–1.94)	
To any vaccine vs. No	12 (1.7)	10 (0.7)	2.41 (1.03–5.59)	2.26 (0.94–5.41)
History of anaphylaxis				
To any drug or food vs. No	5 (0.7)	8 (0.6)	1.24 (0.41–3.81)	
To any vaccine vs. No	4 (0.6)	3 (0.2)	2.66 (0.59–11.90)	
Pre-vaccination medication				
Acetaminophen vs. No	158 (22.5)	232 (16.6)	1.47 (1.17–1.84)	1.23 (1.00–1.60)
NSAID vs. No	16 (2.3)	40 (2.9)	0.79 (0.44–1.42)	
Anti-histamines vs. No	7 (1.0)	4 (0.3)	3.49 (1.02–11.99)	4.38 (1.24–15.49)

Data are presented as numbers (%) of respondents. BNT162b2 #2, the second dose of BNT162b2; ChAdOx1 #1, the first dose of ChAdOx1; 95% CI, 95% confidence interval; NSAID, non-steroid anti-inflammatory drug.

## Data Availability

The data presented in this study are available on request from the corresponding author. The data are not publicly available due to restricted consent.

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
