# Peer review of "Impacts of Side Effects to BNT162b2 and the First Dose of ChAdOx1 Anti-SARS-CoV-2 Vaccination on Work Productivity, the Need for Medical Attention, and Vaccine Acceptance: A Multicenter Survey on Healthcare Workers in Referral Teaching Hospitals in the Republic of Korea"

_vaccines, 2021, doi:10.3390/vaccines9060648_

Round 1

Reviewer 1 Report

This is well-structured and detailed manuscript reporting the results of a survey to assess AEs following vaccination with BNT162b2 and ChAdOx1 anti-SARS-CoV-2 vaccines and their impact on work, medical use, and vaccine acceptance.

I have two main comments to this manuscript:

  1. As one of the aims is to assess the impact of vaccination-related AEs on vaccine acceptance, please describe the relationship between these two variables.
  2. The limitation section should be improved. Please explain how you accounted for recall bias and other limitations (including lack of information on duration of AEs), and explain in which way passive pharmacovigilance efforts can integrate active pharmacovigilance efforts. 

Reviewer 2 Report

The paper is not suitable for publication in the present form. The issue of side effects is important and the number of individuals studied appreciably high therefore I encourage the Authors to improve the shape of the paper making the information provided better documented and much better presented.

The English language should be corrected.

Comments

I recommend using side effects instead of adverse effects, except if a given side effect is potentially harmful the use in the description adverse effect is justified.

l.46

the statement on controversies regarding the ChAdOx1 Astra Zeneca vaccine  should be elucidated by providing frequencies

l.65,

allocation design is not clear, if we compare two groups both should be described according to the parameters they can bias comparisons made. If the groups differ it should be reported and discussed.

  1. 72

It would facilitate reading if distances between the first and second dose with appropriate metrics will be provided and the calendar dates in brackets if needed.

The study flow-chart is too complex, if naming  participated hospitals is important the reason for it should be stated and the differences between the hospitals if present which could affect the study results should be revealed

A lack of information on the presence of side effects after the second jab of Astra Zeneca vaccine does not disqualify the study but it should be clearly stated beginning from the title,  and pertinent paragraph in the abstract section should be rephrased to let people know that the second dose side effects were not registered for some reasons in the AstraZeneca group.

I do not believe that the comparison made between the proportions of side effects registered after the second dose of mRNA vaccine with those registered after the first dose of the Astra Zeneca vaccine is appropriate. The proportions of side effects frequency after the second dose of mRNA may be given separately.

The demographics of the groups should be given rather in the table to be readable and understandable, the present text is difficult to follow.

The side effects should be well described for me it is difficult to understand what does it mean difficulty at work, whether it was workplace difficulty or some individuals were unable to work if so the cause of it should be described. Grading, of the symptoms, must be clearly described and standardized. Whether people reported difficulty to work easily fatigued, had muscular fatigability, tiredness, or simply weakness.  The precise information is crucial. Comparisons made between two vaccines used if made should be illustrated with proportions of symptoms frequency after the first dose. The second dose side effects may be presented separately or in the text by giving simply the proportions. The first dose comparison between the two vaccines used shows a high proportion of side effects when the Astra Zeneca vaccine was used. To make the picture clearer the symptoms should be graded at least for the symptoms which restricted the normal activity or not and those they required medication or not. If the symptoms lasted longer than 48 hrs (as the Authors mentioned) they should be described in some details

The Authors seem to blame the presence of intravascular coagulation as a cause of higher proportions of side-effects in the Astra Zeneca group, however, I believe that this assumption should be much better documented.

Reviewer 3 Report

Exploring the impact of adverse reaction after vaccination with BNT162b2 or Astrazeneca is extremely timely and interesting for the wide public. 

The method is reasonable and well explained. The results appear coherent and clear. The conclusions are fully supported by the results.

I do not recommend severe changes, but only editing to the language to avoid typos.

Round 2

Reviewer 2 Report

In the revised version there are still some drawbacks which I am specified as follows?

  1. In the abstract, the findings of interest should be shortly listed focusing on those which were important and significant. The incidence of side-effects reported as far as I understood the message of the study was interesting being higher among younger people receiving  ChAdOx1 #1vaccine but not in those over 40 yrs. It should be appropriately highlighted in the abstract.   The meaning of the impaired work performance should be defined. The cause of hospitalization should be shown, if there was no predominant cause it should be also mentioned. 5% of hospitalized people after the ChAdOx1 #1vaccine is quite impressive. It should be discussed in some depth.
  2. Side effects reported by individuals receiving ChAdOx1 #1vaccine were not frequent shortly after vaccination, but in this group, there was quite a lot (8.5%)of those they had to request medical assistance including those in emergency departments – the reason for these interventions should be listed and compared to BNT vaccine.

My conclusion sounds as follows;  the study is important in deepening our knowledge in understanding  vaccination associated publicity. However, the impressive proportions of individuals seeking medical assistance after vaccination should be well discussed, Whether reasons for that were rather characteristic for any vaccination using a viral vector vaccine, or they are some peculiarities that have to be identified and appropriately weighted for their risk. For that more thorough information is needed as to the causes of hospitalization and seeking medical assistance. I believe, that in this paper in the present form there are a huge number of information among which the most important ones can be lost. My suggestion is to shorten the paper, to focus the readers on solid data I mean statistically significant and well described, and to discuss them not going too much beyond the facts. If the authors collected information on thrombotic thrombocytopenic manifestation incidence after vaccination, not difficult to obtain for hospitalized or seeking medical assistance patients – it would be valued. 
